# Ecosystem service value evaluation method in a complex ecological environment: A case study of Gansu Province, China

**Xiaojiong Zhao** [1,2,3], **Jian Wang** [1]*, **Junde Su** [4], **Wei Sun** [3]

**1** Northwest Institute of Eco-Environment and Resources, Chinese Academy of Sciences, Lanzhou, China, **2** University of Chinese Academy of Sciences, Beijing, China, **3** Gansu Academy of Eco-environmental Sciences, Lanzhou, China, **4** Gansu Vocational & Technical College of Nonferrous Metallurgy, Jinchang, China

☯ These authors contributed equally to this work.

* wjian@lzb.ac.cn

**Data Availability Statement:** All relevant data are within the manuscript and its Supporting Information files.

## Abstract

The scientific assessment of regional ecosystem service value (ESV) is helpful in developing scientific ecological protection plans and compensation policies. However, an ESV evaluation method that can adapt to the complex and diverse characteristics of the ecological environment has not been established. This study takes Gansu Province in China as an example, fully considering the regional differences in ecosystem service function. Five correction indices for the value equivalent factor per unit area were constructed on a provincial scale, and a regional difference adjustment index for 11 categories of ecosystem services was constructed on a regional scale. In this way, a value evaluation model based on regional differences was established. The results show that in 2015, the total ESV reached 2,239.56 billion yuan in Gansu Province, with ESV gradually increasing from the northeast to the southwest, and the high-value areas of service function being located in Qilian and Longnan Mountains. The forest and grassland ecosystems contributed the most to the ESV. From the perspective of value composition, local climate regulation and biodiversity maintenance functions are the main service functions of Gansu Province. From 2000 to 2015, ESV increased by 3.43 billion yuan in the province. The value of forest and urban ecosystems continued to increase, whereas the value of cultivated land ecosystem continued to decrease. In terms of spatial characteristics of the service value change, the area that experienced value reduction gradually moved from the central part of Gansu Province to the surrounding areas. The evaluation method proposed in this paper provides a relatively comprehensive evaluation scheme for the spatiotemporal dynamic evaluation of ESV in complex ecological environments.

## Introduction

Ecosystems not only provide various raw materials or products directly for human survival, but also have other functions such as regulating climate, reducing pollution, conserving water sources, maintaining soil quality, preventing wind and sand erosion, reducing disasters such as floods and fires, and protecting biodiversity. All ecosystem products and services are

**Funding:** The sources of funding for my study 1. This research was supported by Gansu Youth Science and Technology Fund Program (NO.18JR3RC420), Gansu Soft science project (NO.20CX3ZA002) and Gansu Social Science Planning Project(NO.19YB155). The funding institution of these projects is the Department of science and technology of Gansu Province. 2. The funding for the research was provided by the funders in the research, and the funding are used for data collection, writing and publishing manuscripts, but not for staff salaries.

**Competing interests:** The authors have declared that no competing interests exist.

collectively referred to as ecosystem services (ES) [1,2]. The evaluation of ecosystem service value (ESV) forms the basis of regional ecological construction, ecological protection, ecological work division, and ecological decision-making regarding natural assets, and has become a popular research topic in ecology [3–5]. Since Costanza first quantified the value of global ES in 1997, ESV calculation has increasingly been used as the core basis of ecological asset accounting, thus helping the spatial cognition and sustainable management of national systems in a more intuitive way [5,6]. However, because of the different choices in parameters set by different scholars, the evaluation results of the same ES may vary greatly, and there is a lack of comparability between the ESV obtained through different pricing methods, while a mature pricing method for ESV has not yet been formed internationally [7–10].

At present, research on the evaluation method of ESV can be roughly divided into two categories. The fist is a method based on the service function price per unit area. This method evaluates some key service functions by means of a series of ecological equations, such as food production, soil and water conservation, carbon and oxygen production, and habitat quality [11–14]. The functional value method can accurately measure the extent of some service functions in a region. However, for different service functions, different ecological equations and parameter inputs are often required, and the calculation process is more complicated [3]. Therefore, this method is mostly applied on a small scale, and the implementation cost is high. In addition, when using this method for evaluation, scholars often lack consideration of the ecological background of the study area, and there is no standard in selecting which service functions to evaluate [15]. These shortcomings result in significant uncertainty of the evaluation results, and limitations in the comparison of results. The second is a method based on value equivalent factor per unit area. This method was first proposed by Costanza et al. [5], and divides different land ecosystems and service functions, obtaining the equivalent value based on meta-analysis and the area of each ecosystem, to obtain the regional ESV. Compared with the functional value method, this method evaluates ESV more effectively on a large scale [16] and is widely used in research [3,5,17–19].

However, scholars have found that the evaluation results of the equivalent factor method are valid and reliable only when the equivalent factor accurately reflects the ecological background in the study area [16,20,21]. The equivalent factor proposed by Costanza et al. [5,17] is aimed at global-scale value assessment, which is not consistent with the real ecological situation in China. Xie et al. [18,19] conducted a survey among Chinese ecologists, and put forward an equivalent factor table of ES for China in 2003 and 2008. In 2015, Xie et al. [3] updated and improved the equivalent factor table by adding information obtained from literature and including regional biomass. This table is currently the most scientific and systematic equivalent factor table in China. The equivalent factor table proposed by Xie et al. [3] essentially reflects the average level of the national ecosystem service function. Many more recent studies [14] have shown that the strength of the different service functions is affected by different ecological processes and conditions. For example, organic matter production, gas regulation, and nutrient cycling function [12] is closely related to net primary productivity (NPP); and water supply and regulation function is closely related to rainfall [22], soil erosion [23], habitat quality [13], and the accessibility of recreational sites [24]. Therefore, when the equivalent factor method is used to evaluate the ecological value of a region, the corresponding spatial correction of the equivalent factor is needed [8,25]. At present, scholars only use biomass or NPP to adapt all types of service functions [3,18,19,26], which does not match the real situation. Xie et al. [18] for the first time selected other ecological indicators (rainfall and soil retention) besides NPP to adapt the service function.

Based on the research framework of value equivalent factor per unit area, we adopted the method of meta-analysis and fully used the evaluation results based on the physical quantity

method to determine the average unit area equivalent factor in different ecosystems in Gansu Province. This method avoids or reduces the subjective conjecture easily caused by relying on the experience of experts. Additionally, abundant ecological environment data are used to correct the equivalent factor, thus completing the evaluation of ESV in complex ecological environments. Compared with previous studies [3,18,19,27,28], the evaluation results are more scientific and reasonable. In the evaluation of ESV, the impact of human activities on the ESV, such as bearing capacity, air pollution, groundwater overdraft, and water pollution. is fully considered.

The types of ecosystems are complex and diverse in Gansu Province. The diversity of ecosystem types has caused significant regional differences. However, current research mostly focuses on single or several ecosystems, and only investigate certain ES functions in the ESV in Gansu Province, such as forests [29–31], grassland [32,33], and cultivated land [34]. Considering the complex ecological environment characteristics in Gansu Province, previous studies have not investigated the ESV considering the regional differences in space, and no value evaluation method has been established according to the specific ecological environment in the region.

This study considers the regional differences and the simplicity of the equivalent factor method. In view of the application of the equivalent factors of various ecosystems on a large scale, it is necessary to closely relate the equivalent factors to the national (large) scale and to the actual situation in Gansu Province. Based on a more refined classification of the ecosystem types in Gansu Province, the study added some ecosystem equivalent factors, constructed a revised index, and revised the equivalent factors studied by Xie et al. [3] to form an equivalent factor table that was suitable for the assessment of ESV in Gansu Province. We constructed 11 regional differential adjustment indexes and readjusted the values of different service functions. Finally, we constructed a regional differential value evaluation model to evaluate the change in ESV in Gansu Province from 2000 to 2015. Considering the increasingly severe shortage in and overuse of ecological services in the region, the results of this study can provide a scientific basis for decision support for local governments to formulate more complete ecological compensation policies.

## Materials and methods

### Case study

The case study region is located in northwest China (Fig 1), at the intersection of three major plateaus—the Loess Plateau, the Qinghai Tibet Plateau, and the Inner Mongolia Plateau—and three natural regions—the northwest arid region, the Qinghai Tibet alpine region, and the eastern monsoon region. Gansu Province is a long and narrow region, covering a total land area of 425,800 km$^2$, with complex and diverse geological landforms and climate types. In addition to the marine ecosystem, there are six main land use or cover types, including forests, grasslands, deserts, wetlands, farmland, and urban areas. The Gansu region forms part of China's "two screens and three belts" strategic ecological security barrier policy, which aims to maintain and protect the survival and reproduction of organisms, maintain the natural ecological balance, and guarantee people's livelihoods on the Qinghai Tibet Plateau, the Sichuan–Yunnan Loess Plateau and the north sand belt. It is an important water conservation and supply area in the upper reaches of the Yangtze River and the Yellow River.

### Data sources

**Ecosystem type data.** We used national ecosystem type datasets from the Satellite Application Center of the Ministry of Ecology and Environment and the Chinese Academy of

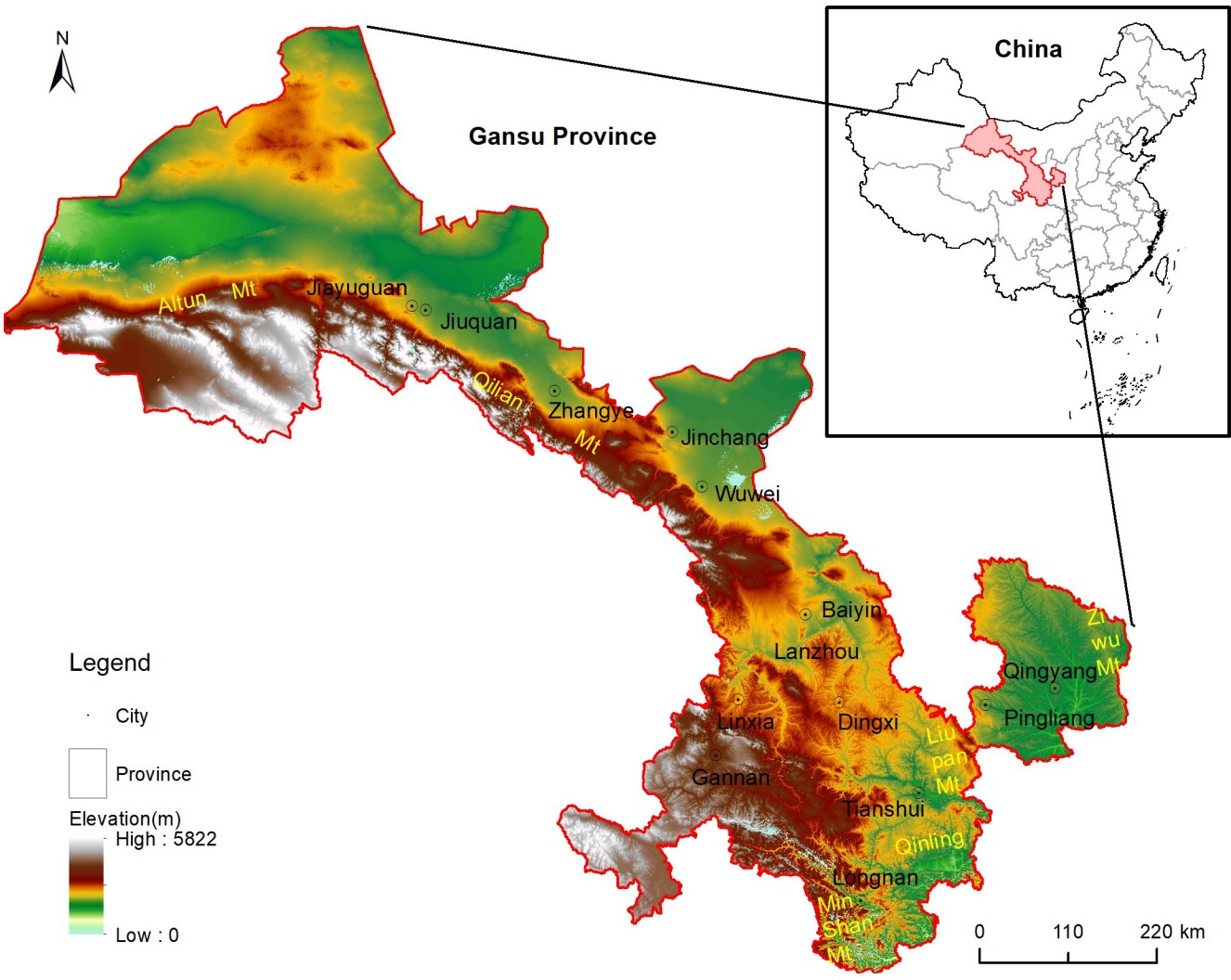

**Fig 1. Location of the study area.**

Sciences, for the periods 2000, 2005, 2010, and 2015, for the classification of ecosystems. The resolution of Landsat TM/ETM images was 30 m, SPOT-5 images was 5 or 2.5 m, Envisat image was 30 m, and HJ-1 images was 30 m. From this data, and according to the study requirements, the ecosystem types were divided into 7 primary types and 21 secondary types in the research area, and a corresponding database was established.

We then used data from 2,508 ground verification points, including 38 different ecosystem types, and integrated these ecosystem types with the ecosystem types obtained through the satellite images, into a final database with 21 ecosystem types, as shown in Fig 2, namely: 1) Deciduous broad-leaved forest; 2) Evergreen coniferous forest; 3) Coniferous broad-leaved mixed forest; 4) Deciduous broad-leaved shrub; 5) Meadow; 6) Grassland; 7) Other grassland; 8) Paddy field; 9) Non-irrigated farmland; 10) Garden land; 11) Herbaceous wetland; 12) Lake; 13) Reservoir; 14) River; 15) Urban green land; 16) Construction land; 17) Bare rock; 18) Bare land; 19) Desert; 20) Saline alkali land; and 21) Glacier. The overall accuracy of the classification was more than 85% [35].

**Meteorological data.**    In this study, the monthly average temperature, precipitation, and sunshine hours from 1981 to 2012 in Gansu Province and its surrounding meteorological

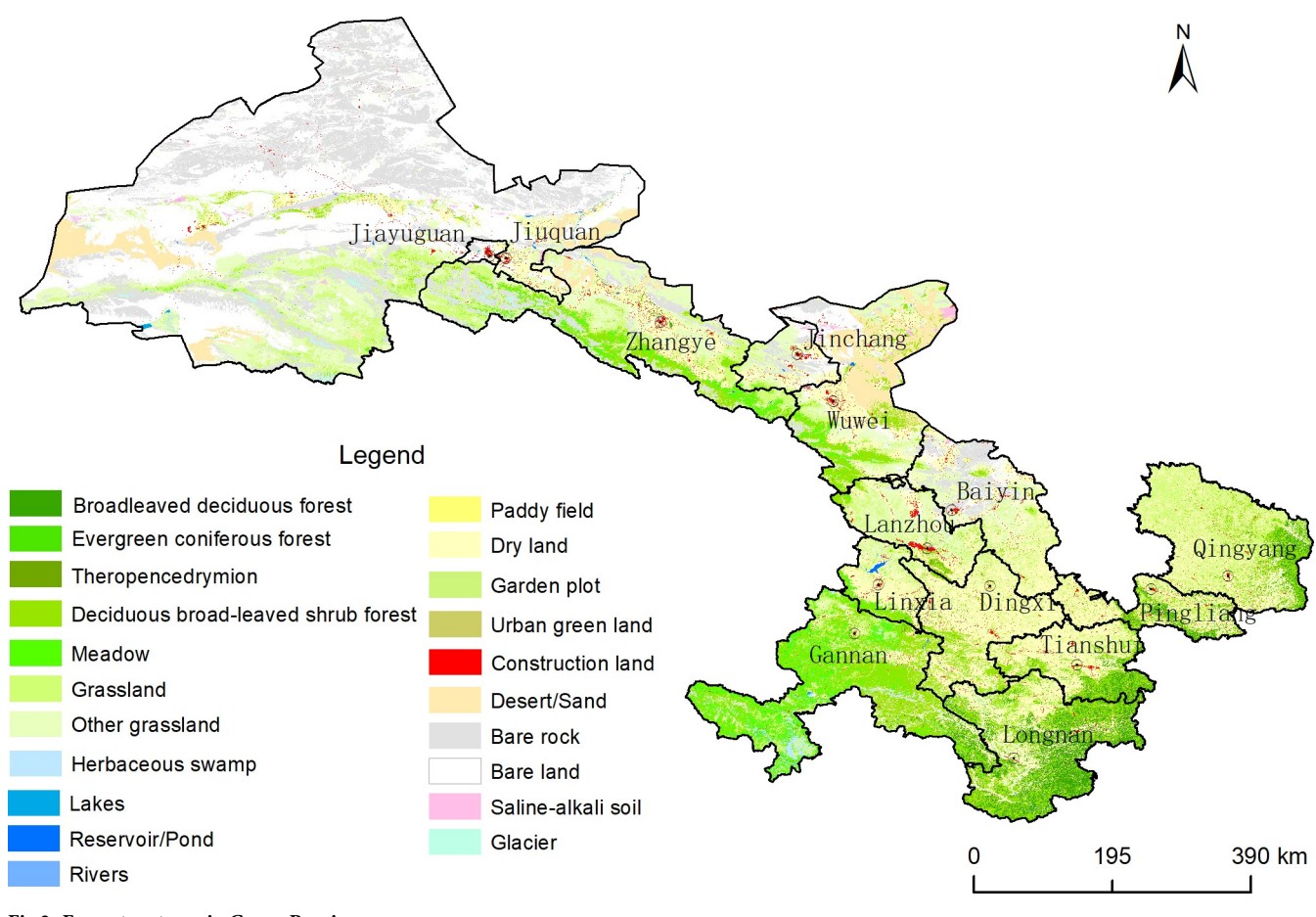

**Fig 2. Ecosystem types in Gansu Province.**

stations were used. The data was obtained from Gansu Meteorological Bureau and China Meteorological Science Data Sharing Service Network (http://cdc.nmic.cn).

**Other geographic data.** The annual average NPP data and the annual average water production data from 2000, 2005, 2010, and 2015 were used in this study, and were obtained from the Satellite Application Center of the Ministry of Ecology and Environment.

**Socioeconomic data.** Social and economic data from 2000 to 2015 were used in this study, and were obtained from the Gansu Province Statistical Yearbook, China Statistical Yearbook, and national agricultural product cost–benefit data. The cultivated land quality data are from the Annual Renewal Evaluation and Monitoring Results of Cultivated Land Quality in Gansu Province (2017), and the grain output of each county is from Gansu Province Rural Yearbook (2000–2014). The monitoring data of atmospheric environmental quality status comes from Gansu Environmental Monitoring Center Station for the period 2015–2018, and the monitoring data of surface water comes from the Bulletin of Environmental Conditions in Gansu Province and the Bulletin of Environmental Conditions in China, for the period 2000–2015.

### Classification of ecosystem service functions

Based on the research results of Costanza et al. [5], de Groot et al. [36], MA [37], and Burkhard et al. [38] on the classification of ES and the characteristics of the ecosystems in Gansu

Province, ES were divided into the following functions: ecological integrity, regulatory services, supply services, and cultural services. Because of the double-counting problem between ecological integrity and other ecosystem services, ecological integrity was, however, not included in the calculation of ESV. Supply services mainly considered crops, livestock, and fresh water; regulation services mainly considered local climate regulation, air quality regulation, groundwater supply, soil conservation, windbreak and sand fixation, and water purification; and cultural services mainly considered entertainment and aesthetic value. Because Gansu is rich in biodiversity, and this forms an important part of the value of ecological resources, the value of biodiversity protection was also included in the value calculation.

## Improved method for value equivalent factor per unit area

**Determination of standard equivalent factor.** The research period in this study is 2000–2015, and the net profit of agricultural products differed each year due to changes in social and economic conditions, and agricultural production technology. The ESV calculated only by the net profit of agricultural products during the study period is not representative. Therefore, in addition to the statistical information obtained (for example, from the Gansu Yearbook, the Second National Agricultural Census in Gansu Province, the Gansu Survey Yearbook, and the compilation of cost and income data of agricultural products in China), the sowing area and net profit of wheat, corn, potato, and oil crops per unit area were obtained, and the average sowing proportion and average net profit were calculated. Based on this, the value of a standard equivalent factor was calculated. The calculation method was as follows:

$$D = S_w \times F_w + S_t \times F_t + S_v \times F_v + S_x \times F_x \tag{1}$$

where $D$ represents the ESV of a standard equivalent factor (yuan $\cdot$ hm$^{-1}$); $S_W$, $S_t$, $S_V$, and $S_X$ represe the sowing area proportion of wheat, corn, potato, and oil to the sowing area of the four crops; and $F_W$, $F_T$, $F_V$, and $F_X$ represent the average net profit of wheat, corn, potato, and oil crops per unit area in Gansu Province (yuan $\cdot$ hm$^{-1}$).

**Value equivalent factor per unit area.** The basic value equivalent of ecosystem service function per unit area (hereinafter referred to as basic equivalent) refers to the annual average value equivalent of various service functions of different ecosystem types per unit area. Previous studies on equivalence factors [3,18,19] are based on the annual average value on a national scale, and have a rough classification of ecosystem types, which cannot meet the need of the refinement of ecosystem classification, nor precisely reflect the difference in service function among ecosystem types. Therefore, in this study, the average value equivalent factor per unit area of different ecosystems was determined, by following the calculation process described below.

1. For the types of ecosystems and the corresponding ecosystem service types in Gansu Province (in cases where there was an equivalent factor in the equivalence factor table of Xie et al. [3]), the national average value equivalence factor was used. On this basis, by constructing the correction coefficient, it was converted into the average value equivalence factor of ecosystem services functions, such as fresh water supply, local climate regulation, entertainment and aesthetic value, air quality regulation, and water purification.

2. Relevant international literature, such as publications by Elsevier, Springer Nature, Wiley, and the Chinese How Net database, was searched. We inputted retrieval words such as Gansu Province, the names of each basin and city in Gansu Province, Qilian Mountain, and Gannan Plateau, to obtain research results on ecosystem service value calculated by ecosystem service function quantity in Gansu Province. If, in the future, there are many

3. We prioritized the collection and sorting of domestic published research results of ecosystem service value calculated by ecosystem service function quantity. The average of selected ESV, and thereafter the proportion with standard equivalents, should be calculated, so as to convert them into the average value equivalent factor of ecosystem service function, as the basic equivalent of the ecosystem service function, which is used to determine the value equivalent of ecosystem service functions, such as garden land, shrub land, forest land, and swamp wetland.

4. If no relevant research results for Gansu Province can be found, relevant research results for other regions in China should be collected. ESV per unit area of ecosystem should be calculated, and then compared with the standard equivalent value. It can be converted into an average value equivalent factor of ecosystem service function by constructing a correction coefficient, as the basic equivalent of the ecosystem service function, which is used to determine the value equivalent of some ecosystem service functions, such as lakes, reservoirs, saline alkali land, and urban green space.

5. There are great differences between the ecosystem service functions in Gansu Province and those of the entire country. Therefore, in this study, the value equivalence factor was localized and calibrated by calculating the ecosystem service function quantity per unit area, such as the biodiversity maintenance value of forest land, grassland, wetlands, and desert ecosystems, and the crop supply service of paddy fields and dry land.

6. If there is no ecosystem service function value listed in directly corresponding documents in the secondary classification of ecosystem, and it is therefore difficult to calculate the ESV, refer to the equivalent factors listed by Xie et al. [3], as they were determined by experts' experience. Transform them into the average value equivalent factors of the ecosystem service function in Gansu Province through use of the correction coefficient, as the basic equivalent of the ecosystem service function, such as local climate regulation and air quality regulation of the secondary types of desert and wetland ecosystems, and soil conservation services and water purification services of the second level of desert ecosystems.

Through the above six steps, the value of the main ecosystem types for a certain ecosystem service function per unit area in Gansu Province can be obtained, by referring to relevant literature or doing calculations, as shown in Table 1.

**Correction index of value equivalent factor per unit area.**

1. Crop supply correction index (*N*)

The calculation method of crop supply correction index is as follows:

$$N = y/Y \tag{2}$$

where $y$ is the average output per unit area in Gansu Province, and $Y$ is the national average output per unit area.

The calculation of y is based on the following formula: $y$ = (average yield per unit area of wheat/average yield per unit area of wheat) × sowing proportion of wheat + (average yield per unit area of corn/average yield per unit area of corn) × sowing proportion of corn + (average yield per unit area of potato/average yield per unit area of potato) × sowing proportion of

**Table 1. Calculation basis of value equivalent factor per unit area in Gansu Province.**

| Ecosystem service function | Calculation basis, literature source | The Value per unit area calculated in this study/ actual estimated value of existing research used |
|---|---|---|
| **Livestock supply** | Maqu grassland [39], 1980s; | 498.44; |
| | Shandan Racecourse [40], 2010 | 170.72; |
| | Etokqianqi grassland [41], 2014 | 1,883.23; |
| **Fresh water supply** | The fresh water supply service of the lakes in Gansu Province (large and small Sugan lakes) is multiplied by the water price; | 4,639.97; |
| | The average annual water supply volume, reservoir area and water price of large and medium-sized reservoirs and dams with statistical data in Gansu Province are used to calculate the unit area value of fresh water supply; | 14,072.60; |
| | Wetlands in Jilin Province (permanent and seasonal rivers) [42], 2013; | 1,2510.00; |
| | Chinese Desert [43]; | 265.69; |
| | Desert equivalent is adopted for fresh water supply service of saline alkali land (no research on fresh water supply of saline alkali land is retrieved). | 265.69; |
| **Local climate regulation** | China Desert [43]; | 1,810.37; |
| | Gansu tea garden, carbon sequestration [44], 2011; | 268.00; |
| | Carbon sequestration and oxygen release of urban green space in Qingdao [45], 2015; | 4,439.40, 17,630.61; |
| | Carbon sequestration, oxygen release and heat island mitigation of urban green space in Jinan [46], 2009; | 4,908.69, 5,785.94, 1,268.78; |
| | Zhengzhou green space carbon sequestration, oxygen release and temperature reduction [47], 2003–2013; | 6,496.37, 10,313.77, 42,108.85; |
| | Bush carbon sequestration and oxygen release in Gansu Province [48], 2015; | 8,076.98; |
| | Carbon sequestration and oxygen release from deserts in China [49], 2004. | 306.00, 281.88; |
| **Air quality regulation** | Greenland, Zhengzhou [47], 2003–2013; | 2,064.53; |
| | Jinan urban green space [46], 2009; | 1,406.39; |
| | Qingdao urban green space [45], 2015; | 5,216.62; |
| **Groundwater recharge** | Farmland irrigation area in Bayin River Basin, Qinghai Province [50]; | 358.52; |
| | GuizhouHuahai wetland (Lake) [51], 2010; | 7,125.52; |
| | KuTang, Jilin Province [52], 2014; | 7,508.12; |
| | Jilin River [49] (Cui et al., 2017), 2014; | 9,265.23; |
| | Jilin herbaceousswamp [52], 2014; | 3,788.73; |
| | Ice and snow melt water supply in Hexi Corridor [53], 2003; | 22.77; |
| | Melting water supply of glaciers in the middle reaches of Heihe River [54], 1987–2000. | 7.44; |
| **Soil conservation** | China Desert [49], 2004; | 177.00; |
| | Gansu thicket [48], 20092015; | 1,353.39; |
| | Shenzhen urban green space [55], 2015; | 1,039.50; |
| | Jinan urban green space [46], 2009; | 1,979.07; |
| | Beijing Garden [56], 2004; | 7,987.86; |
| | Gansu tea garden [44], 2011; | 33.00; |
| **Windbreak and sand fixation** | Etuokeqian grassland [41], 2014; | 3,817.69; |
| | Desert of Maduo County [57], 2011, 2014; | 343.68; |
| | Abihu Gobi [58], 2000–2015; | 248.91; |
| | ABI Lake bare rock land [57], 2000–2015; | 38.19; |
| | ABI lake saline alkali land [58], 2000–2015; | 388.58; |
| | Dry land of Ebinur Lake [58], 2000–2015; | 3,313.41; |
| | JingdianIrrigation District forest [59], 2016; | 9,678.40; |

(*Continued*)

**Table 1.** (Continued)

| Ecosystem service function | Calculation basis, literature source | The Value per unit area calculated in this study/ actual estimated value of existing research used |
|---|---|---|
| **Water purification** | Forest land of Qilian Mountain Nature Reserve [60], 2008; | 2,959.45; |
| | Shrubbery in Qilian Mountain Nature Reserve [60], 2008; | 2,642.03; |
| | Bailongjiang nature reserve forest [61], 2005; | 3,873.79; |
| | Coastal saline alkali land [62], 2000, 2011 | 2,904.51; |
| | Baisha reservoir, Henan Province [63], 2018; | 10,596.60; |
| | Lishimen reservoir, Zhejiang Province [64], 2018; | 9,281.33; |
| | Six key reservoirs in Zhejiang Province [65], 2011; | 19,963.12; |
| **Aesthetic entertainment value** | Zhangye Heihe wetland [66], 2012; | 10,102.95; |
| | Beijing Garden [56], 2004; | 16.14; |
| | Shenzhenurban green space [55], 2015; | 907.52; |
| | Bosten Lake [67], 2012; | 8,440.00; |
| **Biodiversity maintenance value** | Gansu broad leaved forest, 2015; | 25,033.95; |
| | Coniferous forest, 2015; | 7,077.54; |
| | Mixed coniferous and broad leaved forest, 2015; | 18,895.23; |
| | Gansu shrub, 2015; | 2,740.60; |
| | Gansu meadow, 2015; | 18,911.41; |
| | Gansu Grassland, 2015; | 9,225.15; |
| | Other grasslands in Gansu, 2015; | 6,309.70; |
| | Gansu Lake, 2015; | 36.69; |
| | Gansu reservoir, 2015; | 73.41; |
| | Gansu River, 2015; | 267.28; |
| | Gansu herbaceous swamp, 2015; | 4,233.29; |
| | Gansu desert, 2015; | 166.02; |
| | Gansu bare rock, 2015; | 285.89; |
| | Gansu saline alkali land, 2015; | 112.61; |
| | Gansu bare soil, 2015; | 174.55; |

potato + (average yield per unit area of oil/average yield per unit area of oil) × sowing proportion of oil.

The calculation method of $Y$ is the same as that of $y$.

2. Fresh water supply correction index ($D$)

The fresh water supply correction index is calculated as follows:

$$D = w/W \qquad (3)$$

where $w$ is the average water supply per unit area in Gansu Province (10,000 m³), and $W$ is the average water supply per unit area in China (10,000 m³). The water supply data comes from the Water Resources Bulletins (2000–2015) for Gansu Province and China.

3. Air quality regulation correction index (K)

The air quality regulation correction index is calculated as follows:

$$K = a/A \qquad (4)$$

where $a$ is the average proportion of air quality standards of prefecture level cities in Gansu Province, and $A$ is the average proportion of air quality standards of prefecture level cities in

China. The air quality standards data come from the Environmental Quality Bulletins (2000–2015) for Gansu Province and China.

4. Water purification correction index ($S$)

The calculation method of water purification correction index is as follows:

$$S = q/Q \tag{5}$$

where $q$ is the average length proportion of class I–III water reach in Gansu Province, and $Q$ is the average length proportion of class I–III water reach in China. The length data of water quality reach is from the Water Resources Bulletins (2000–2015) for Gansu Province and China.

5. Entertainment and aesthetic value correction index ($Y$)

The calculation method of entertainment and aesthetics value is calculated as follows:

$$Y = r/R \tag{6}$$

where $r$ is the average tourism revenue per unit area in Gansu Province, and $R$ is the average tourism revenue per unit area in China. The tourism revenue data comes from the Statistical Yearbooks (2000–2015) for Gansu Province and China.

## Regional difference adjustment index

1. Crop supply regulation index ($A1$)

The crop supply regulation index is calculated as follows:

$$A1i = ai/A \quad (i \text{ is } 1, 2) \tag{7}$$

where $ai$ is the average yield per unit area in Gansu Province, $A$ is the average yield per unit area in Gansu Province, the calculation method of $ai$ and $A$ is the same as in Eq (1), $a1$ is the high-yield area, and $a2$ is the low-yield area.

According to Cheng [68], there are obvious spatial differences in grain production of cultivated land in Gansu Province. Previous studies [69] found that the correlation between cultivated land quality and land use is relatively high, with the correlation coefficient reaching 0.874. First, this was reflected in the cultivated land quality level (land use level) of each county as referred to in related studies [70] on the spatial distribution of land use level in 2015 in Gansu Province. Second, in terms of proportion of cultivated land use level to the total cultivated land area of each county in Gansu Province, 27% of the counties were classified as high-yield areas, and the rest were classified as low-yield areas. Finally, the average yield per unit area was calculated in high- and low-yield areas of each county, as well as in the province as a whole (refer to the calculation result of Eq (1)). The average yield in the high- and low-yield areas was compared with the average yield per unit area in Gansu Province, and the regulation index of crop supply in high- and low-yield areas was obtained.

2. Livestock supply adjustment index ($A2$)

The livestock supply adjustment index is calculated as follows:

$$A2i = bi/b3 \quad (i \text{ is } 1, 2, 3) \tag{8}$$

where $bi$ is the average livestock carrying capacity of each area; $b1$ is the average livestock carrying capacity in agricultural areas; $b2$ is the average livestock carrying capacity in semi-pastoral areas; and $b3$ is the average livestock carrying capacity in pastoral areas.

Gansu Province is divided into pastoral, semi-pastoral, and agricultural areas because of the great difference in livestock supply capacity between pastoral and agricultural areas. Previous studies [71] have calculated the livestock carrying capacity in agricultural and pastoral areas of Gansu Province, and found that the livestock carrying capacity of agricultural areas was 0.85 times that of pastoral areas. The average of livestock carrying capacity in agricultural and pastoral areas was considered the livestock carrying capacity in the semi-pastoral areas.

3. Fresh water supply regulation index ($A3$)

The fresh water supply regulation index was calculated as follows:

$$A3i = ci/c1 \quad (i \text{ is } 1, 2, 3) \tag{9}$$

where $ci$ is the average water yield per unit area of each area; $c1$ is the average water yield per unit area in the water rich areas; $c2$ is the average water yield per unit area in the water poor areas; and $c3$ is the average water yield per unit area in the dry areas.

The distance between the east and west, and the north and south is large in Gansu Province, and precipitation decreases from southeast to northwest due to the influence of water vapor and terrain. According to research [72], Gansu Province is divided into abundant water areas (Liupanshan–Longshan area, Longnan Mountain area, Gannan Plateau, and Qilian Mountain Area), water poor areas (Longdong, Longdong–Loess Plateau area, north of Lanzhou area), and dry areas (Hexi Corridor, Beishan Mountain area, and the desert area bounded by the Qilian Mountain foot). Using the water yield module in the Integrated Valuation of Ecosystem Services and Tradeoffs (InVEST) model, the average multi-year water yield of the water rich areas, water deficient areas, and dry areas was calculated for 2000–2015. The water yield per unit area in the water deficient areas was 0.68 times that in the water rich areas, and the water yield per unit area in the dry areas was 0.08 times that in the water rich areas.

4. Local climate regulation index (A4)

The local climate regulation index was calculated as follows:

$$A4i = di/d2 \quad (i \text{ is } 1, 2, 3) \tag{10}$$

where $di$ is the average NPP of each partition; $d1$ is the NPP value of the high adjustment area; $d2$ is the mean value of NPP in the middle regulation area; and $d3$ is the NPP value of the low regulation area.

A large amount of observation data analysis shows that a change in surface vegetation may have a significant impact on local and regional climate by changing surface attributes such as surface albedo, roughness, and soil moisture [73–76]. The higher the area of vegetation NPP, the stronger the function of climate adjustment. Therefore, the value of NPP is used to measure regional differences in climate regulation. According to the spatial distribution characteristics of NPP and the boundaries of townships, Gansu Province is divided into high regulation area, middle regulation area, and low regulation area. The average value of NPP in the three regulation areas was calculated, and the ratio of NPP between the high regulation area and the middle adjustment area was taken as the adjustment index in the high adjustment area. The average value ratio of NPP between the low adjustment area and the median adjustment area was used as the adjustment index of the low value area.

5. Air quality regulation index ($A5$)

The air quality regulation index was calculated as follows:

$$A5i = ei/e2 \quad (i \text{ is } 1, 2, 3) \tag{11}$$

where $ei$ is the average vegetation coverage in each area; $e1$ is the average vegetation coverage in the area with good air quality; $e2$ is the average vegetation coverage in the area with average air quality; and $e3$ is the average vegetation coverage in the area with poor air quality.

Generally, the better the air quality in a region, the greater the air quality regulation service function. According to the 2015 Environmental Quality Bulletin in Gansu Province, $PM_{10}$ and $PM_{2.5}$ were the main air pollutants. Only one of the 14 cities and prefectures has reached the secondary standard of ambient air quality, so the concentration of pollutants was taken as the index to measure the level of air quality regulation function. In this study, $PM_{10}$ and $PM_{2.5}$ concentration monitoring data were selected at 111 provincial monitoring points in Gansu Province, and through Kriging interpolation, the spatial distribution of $PM_{10}$ and $PM_{2.5}$ concentration was obtained in the whole province, which was divided into three zones: the area meeting the secondary quality standard was classified as the area with the best air quality, indicating that the area had the highest air quality regulation function; the other two areas were demarcated according to pollutant concentration. Vegetation coverage is closely related to air purification function. In this study, the ratio of the average vegetation coverage in the three regions was used as the air quality regulation index.

6. Groundwater recharge regulation index ($A6$)

The groundwater recharge index was calculated as follows:

$$A6i = fi/f1 \quad (i \text{ is } 1, 2) \tag{12}$$

where $fi$ is the ratio of actual exploitation amount and exploitable amount of groundwater in each zone; $f1$ is the ratio of actual exploitation amount and exploitable amount of groundwater in heavily mined areas; and $f2$ is the ratio of actual exploitation amount and exploitable amount of groundwater in areas that are not heavily mined, assuming that the actual exploitation amount and exploitable amount of groundwater in those areas are balanced, with the ratio set as 1.

The overexploitation of groundwater results in drainage of the aquifer, a decrease in groundwater level, the formation of a funnel, and land subsidence. Therefore, when rapid development exceeds the resource stock and environmental capacity, the value of the groundwater ecosystem will inevitably continue to appreciate. Different regions have different needs for groundwater recharge function, resulting in different values. To reflect the regional differences in groundwater recharge regulation function value, we used groundwater in heavily mined areas and areas that are not heavily mined to measure the regional differences in groundwater recharge function. Groundwater in heavily mined areas has a higher groundwater recharge value than in areas that are not heavily mined. According to the delimitation results of groundwater in heavily mined areas in Gansu Province [77], there are 46 heavily mined areas, involving 32 counties. The ratio of the actual and exploitable groundwater in the heavily mined area is used as the groundwater supply regulation index.

7. Soil conservation regulation index ($A7$)

The soil conservation regulation index was calculated as follows:

$$A7i = gi/g \quad (i \text{ is } 1, 2) \tag{13}$$

where $gi$ is the average erosion modulus of each area; $g1$ is the average erosion modulus of the key prevention area; $g2$ is the average erosion modulus of the key control area; and $g$ is the allowable amount of soil erosion.

Gansu Province is located at the junction of three plateaus, and its soil conservation functions vary substantially in different areas. In terms of soil and water loss in the key prevention

and key governance areas, there is better vegetation, less soil and water loss, and stronger soil conservation functions in the key prevention areas, but low forest and grass coverage, a fragile ecological environment, and extensive soil and water loss in the key governance area. Therefore, the province is divided into two zones according to the range of the key prevention and governance areas. According to classification standards for soil erosion, the allowable amount of soil and water loss in the northwest Loess Plateau is 1,000 t/km$^2$. Based on the ratio of the average erosion modulus and the allowable amount of soil and water loss in the two zones, the adjustment index of soil conservation was constructed.

8. Regulation index of windbreak and sand fixation (*A8*)

The windbreak and sand fixation regulation index was calculated as follows:

$$A8i = hi/h1 \quad (i \text{ is } 1, 2) \tag{14}$$

where $hi$ is the amount of windbreak and sand fixation in each zone; $h1$ is the amount of windbreak and sand fixation in the service area of windbreak and sand fixation; and $h2$ is the amount of windbreak and sand fixation in other areas.

The Hexi Corridor in the north of Gansu Province, and the surrounding county of Qingyang City is located in the Gobi Desert area. Therefore, this area is classified as a service area for windbreak and sand fixation, whereas other areas are not considered to have that function.

9. Water purification regulation index (*A9*)

The water purification regulation index was calculated as follows:

$$A9i = ji/j2 \quad (i \text{ is } 1, 2) \tag{15}$$

where $ji$ is the target proportion of water quality in each zone; $j1$ is the length proportion of class II and above water reaches in the high water purification area; and $j2$ is the length proportion of class III and below water reaches in the low water purification area.

Xie et al. [78] highlighted that, as the pollution of rivers and lakes is becoming more serious, the water quality regulation function of rivers is becoming lower, and rivers and lakes almost become an area of accumulation of waste. Therefore, the water quality of the reach is closely related to its water purification function. If the water quality of this area is significantly better than that of other areas, the water purification function of this area will be of considerable importance. In this study, the water quality objectives of 236 monitoring sections of the Rivers were used to measure the water purification function of the region, and the water quality objectives of each county were quantified. If the water quality objectives of class I, or class II, or class II water and class III water reaches can be achieved simultaneously, the water purification function of the county is considered to be high. Additionally, the length proportion of the class II and class III water reaches and below are calculated. The length proportion of the class II water reach to the class III water reach and below is taken as the water purification regulation index in the high water purification area.

10. Entertainment aesthetics value adjustment index (*A10*)

The value adjustment index of entertainment and aesthetics was calculated as follows:

$$A10i = ki \,(i \text{ is } 1, 2, 3) \tag{16}$$

where $ki$ is the value adjustment index of entertainment aesthetics in each zone; $k1$ is the value adjustment index of entertainment and aesthetics in key tourist areas; $k2$ is the value adjustment index of entertainment and aesthetics in general tourist areas; and $k3$ is the value adjustment index of entertainment and aesthetics in other regions.

Entertainment value refers to the value obtained by tourists when they are engaged in tourism activities in an ecotourism scenic spot, which is the sum of the value used by direct recreation and the non-use value possessed by resources; aesthetic value refers to the pleasure value brought by natural ecosystems to people's aesthetic perception of the natural and cultural landscape, and the value of its own objective aesthetic attributes is referred to as the non-use value. If people cannot reach an area that can bring recreational and pleasure value to people, it is considered that the area cannot provide the service function or temporarily does not have the function. Based on this, we first determined that the core area and buffer area of a nature reserve cannot provide this service temporarily. Other natural protected places such as forest parks, geoparks, scenic areas, and wetland areas are key tourism areas, which provide the highest entertainment and aesthetic value. Second, the whole tourism county (and not only the key tourism areas) is generally regarded as the tourism area, with the remaining areas having the lowest entertainment and aesthetic value. The adjustment indexes of the different regions are assigned by expert judgment.

11. Biodiversity maintenance value regulation index ($a11$)

The value adjustment index of biodiversity maintenance was calculated as follows:

$$A11i = li/l1 \quad (i \text{ is } 1, 2) \tag{17}$$

where is the habitat quality index of each region; $l_1$ is the habitat quality index of the priority area for biodiversity conservation; and $l_2$ is the habitat quality index of other regions.

According to the conservation plan for biodiversity priority areas in Gansu Province, there are seven biodiversity priority areas in the province. This study considered that the areas located in the priority areas had the highest biodiversity maintenance value, followed by other areas, and the province was therefore divided into two areas. To determine the adjustment index of the different regions, we used the habitat quality module of the InVEST model to calculate the habitat quality index of the different regions [79], and determined the adjustment index by comparing the size of the habitat quality index of the two regions.

Through the above methods, the value equivalent factor table per unit area was established for Gansu Province (Table 2).

## Value evaluation model based on regional differences

Different geomorphic types will affect the distribution of light, heat, water, and soil types in the region [80]. Similarly, different regional ecological environments, degree of ecological protection, intensity of ecological demand for different land use types, and implementation of local policies will affect the benefits human beings derive from the ecosystem, thus affecting the regional differences and divisions of ESV. The ArcGIS spatial analysis tool was used to grid Gansu Province, and a complete grid of 1 km × 1 km was extracted. Based on the calculation of the ESV of the grid unit, the total ESV ($V$) was calculated using the following equation (Fig 3):

$$V = \sum_{i=1}^{c} V_c \tag{18}$$

where $V_c$ is the value of ecosystem service function $c$; $c$ is ecosystem service function, and the value is between 1 and 11.

$$V_c = \sum_{i=1}^{n} \sum_{j=1}^{m} D \cdot F_m \cdot A_c \cdot S_m \tag{19}$$

where $D$ is the standard equivalent factor; $F_m$ is the value equivalent factor per unit area of ecosystem type m; $A_c$ is the regional difference adjustment index of ecosystem service function $c$; $S_m$ is the area of ecosystem type $m$ (km²); and $n$ is the grid number.

**Table 2. Value equivalent factors per unit area in Gansu Province.**

| Primary types | Secondary types | Supply services | | | Regulatory services | | | | | | Cultural Services | - |
|---|---|---|---|---|---|---|---|---|---|---|---|---|
| | | A | B | C | D | E | F | G | H | I | J | K |
| Cultivated land | 1 | 1.07 | 0.00 | -2.63 | 0.57 | 0.14 | 0.14 | 0.01 | 0.00 | 0.18 | 0.07 | 0.09 |
| | 2 | 0.67 | 0.00 | 0.01 | 0.36 | 0.08 | 0.00 | 1.03 | 1.34 | 0.11 | 0.04 | 0.06 |
| | 3 | 5.82 | 0.00 | 0.12 | 0.11 | 1.54 | 0.00 | 1.62 | 1.06 | 2.05 | 0.01 | 0.48 |
| Forest | 4 | 0.22 | 0.00 | 0.13 | 5.07 | 1.19 | 0.00 | 2.06 | 0.00 | 1.58 | 0.28 | 2.85 |
| | 5 | 0.31 | 0.00 | 0.17 | 7.03 | 1.59 | 0.00 | 2.86 | 0.00 | 2.11 | 0.39 | 7.61 |
| | 6 | 0.29 | 0.00 | 0.16 | 6.50 | 1.54 | 0.00 | 2.65 | 3.84 | 2.05 | 0.36 | 10.09 |
| | 7 | 0.19 | 0.00 | 0.10 | 3.25 | 1.44 | 0.00 | 0.55 | 0.97 | 1.36 | 0.23 | 1.10 |
| Grassland | 8 | 0.22 | 0.20 | 0.08 | 3.02 | 0.80 | 0.00 | 1.39 | 0.60 | 1.06 | 0.43 | 7.62 |
| | 9 | 0.10 | 0.07 | 0.04 | 1.34 | 0.35 | 0.00 | 2.00 | 1.54 | 0.47 | 0.19 | 3.72 |
| | 10 | 0.38 | 0.07 | 0.15 | 5.21 | 1.38 | 0.00 | 2.40 | 0.60 | 1.82 | 0.33 | 2.54 |
| Wetland | 11 | 0.00 | 0.00 | 1.87 | 2.29 | 2.88 | 2.87 | 0.93 | 0.00 | 4.25 | 0.64 | 0.01 |
| | 12 | 0.00 | 0.00 | 5.67 | 2.29 | 2.88 | 3.03 | 0.93 | 0.00 | 5.67 | 0.64 | 0.03 |
| | 13 | 0.00 | 0.00 | 5.04 | 2.29 | 2.88 | 3.73 | 0.93 | 0.00 | 5.88 | 0.64 | 0.11 |
| | 14 | 0.51 | 0.00 | 2.59 | 3.60 | 2.88 | 1.53 | 2.31 | 0.14 | 3.82 | 4.11 | 1.71 |
| Cities | 15 | 0.00 | 0.00 | 0.00 | 0.00 | 0.10 | 0.00 | 0.00 | 0.10 | 0.11 | 0.01 | 0.00 |
| | 16 | 0.00 | 0.00 | -0.23 | 5.54 | 0.70 | 0.00 | 0.61 | 1.06 | 1.35 | 0.37 | 0.79 |
| Desert | 17 | 0.00 | 0.00 | 0.11 | 0.73 | 0.08 | 0.00 | 0.07 | 0.10 | 0.11 | 0.02 | 0.06 |
| | 18 | 0.00 | 0.00 | 0.00 | 0.24 | 0.08 | 0.00 | 0.07 | 0.02 | 0.11 | 0.00 | 0.12 |
| | 19 | 0.00 | 0.00 | 0.11 | 0.24 | 0.08 | 0.00 | 0.07 | 0.16 | 1.24 | 0.00 | 0.05 |
| | 20 | 0.00 | 0.00 | 0.00 | 0.24 | 0.08 | 0.00 | 0.07 | 0.10 | 0.11 | 0.00 | 0.07 |
| Glacier | 21 | 0.00 | 0.00 | 1.02 | 0.54 | 0.13 | 0.01 | 0.00 | 0.00 | 0.17 | 0.03 | 0.01 |

1 Paddy field; 2 Non irrigated farmland; 3 Garden land; 4 Deciduous broad-leaved forest; 5 Evergreen coniferous forest; 6 Coniferous broad-leaved mixed forest; 7 Deciduous broad-leaved shrub; 8 Meadow; 9 Grassland; 10 Other grassland; 11 Lake; 12 Reservoir; 13 River; 14 Herbaceous wetland; 15 Construction land;16 Urban green land; 17 Desert; 18 Bare rock; 19 Saline alkali land;20 Bare land; 21 Glacier.

A: Crops supply; B: livestock supply; C: Fresh water supply; D: Local climate regulation; E: Air quality regulation; F: Groundwater supply; G: Soil conservation; H: Windbreak and sand fixation; I: Water purification; J: Entertainment aesthetic value; K: Biodiversity maintenance value.

## Results

### Analysis of the change in ecosystem in Gansu Province

Desert is the largest ecosystem type in Gansu Province (Table 3), followed by grassland, arable land, and forest. Both the glacier and wetland ecosystems account for a small part. Desert is mainly found in the north of Gansu; grasslands are mainly distributed in the Gannan Plateau in central and eastern Gansu; cultivated land is mainly distributed in the central area of Gansu and Hexi Corridor; and forests are mainly found in the Longnan, Ziwuling, and Qilian Mountains. In terms of changes in the ecosystem, the forest, grassland, and urban ecosystems have been increasing continuously in the past 15 y. The cultivated land ecosystem has been continuously decreasing. Although the wetland and glacier ecosystems have been increasing in the past 15 y, they have been decreasing over the longer term. The desert ecosystem has been increasing in general.

### Overall evaluation of ESV in Gansu Province

In general, ESV decreases from south to north, and from east to west in Gansu Province (Fig 4), which is consistent with the spatial distribution of forest, grassland, and desert ecosystems. There are adjoining desert areas north of Gansu Province, the area with the lowest ESV.

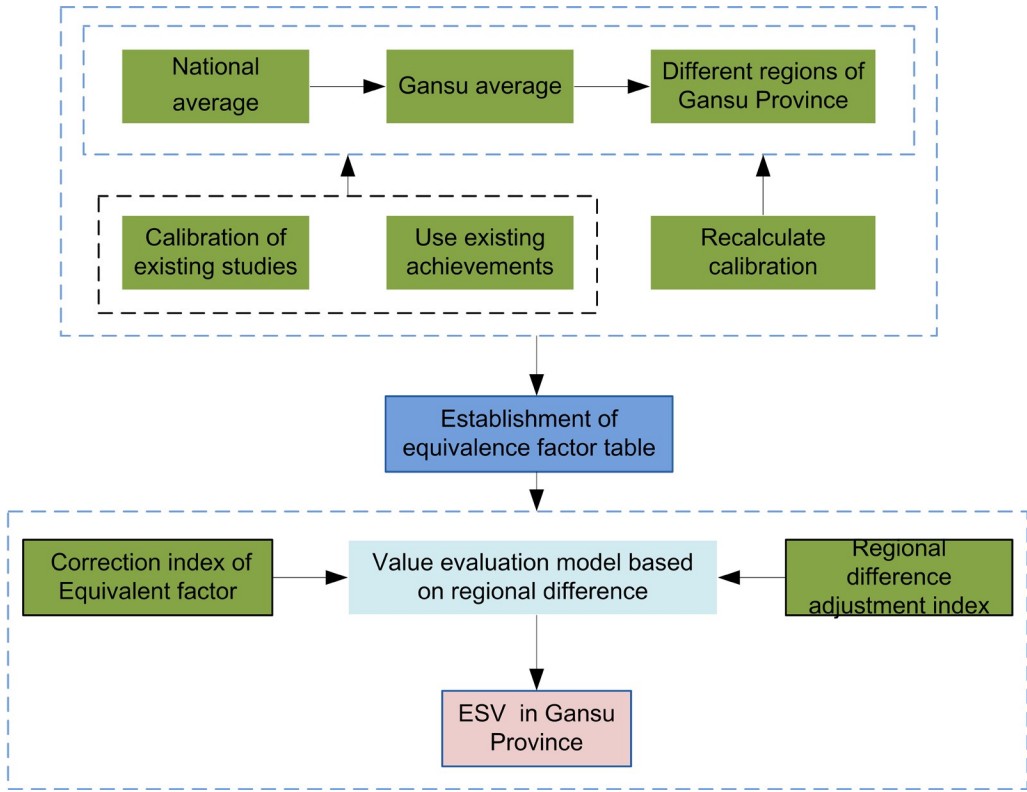

**Fig 3. Flow chart of ecosystem service value evaluation model based on regional differences.**

However, the ecological environment is relatively healthy in the south of Gansu Province, with high vegetation coverage and ESV. In the center-east of Gansu Province, where human activities are most frequent, inducing relatively high disturbance on the natural ecological environment, the ESV is relatively low.

The value of local climate regulation and biodiversity maintenance (Table 4) is much higher than that of other service functions, and constitutes the main contributor to ESV in Gansu Province. In terms of the change in each service function value, the value of soil conservation, windbreak and sand fixation, and biodiversity protection has been increasing continuously in the past 15 y, whereas the supply value of crops has been decreasing continuously. The supply value of fresh water, local climate regulation, air quality regulation, water purification function, groundwater supply, and entertainment and aesthetic value show a fluctuating change state and an overall decreasing trend.

**Table 3. Change of ecosystem pattern from 2000 to 2015 in Gansu Province (km².).**

| Ecosystem types | 2000 | 2005 | 2010 | 2015 |
|---|---|---|---|---|
| Forest | 54,372.71 | 55,118.01 | 56,178.19 | 56,228.06 |
| Grassland | 120,419.34 | 122,061.74 | 124,629.60 | 124,739.27 |
| Wetland | 2,624.82 | 2,732.48 | 2,440.76 | 2,545.59 |
| Cultivated land | 76,454.39 | 74,362.82 | 68,743.14 | 68,328.45 |
| Cities | 3,438.61 | 3,723.99 | 4,071.09 | 4,624.28 |
| Desert | 167,223.90 | 166,546.96 | 168,485.42 | 168,076.37 |
| Glacier | 909.06 | 896.83 | 894.63 | 900.81 |

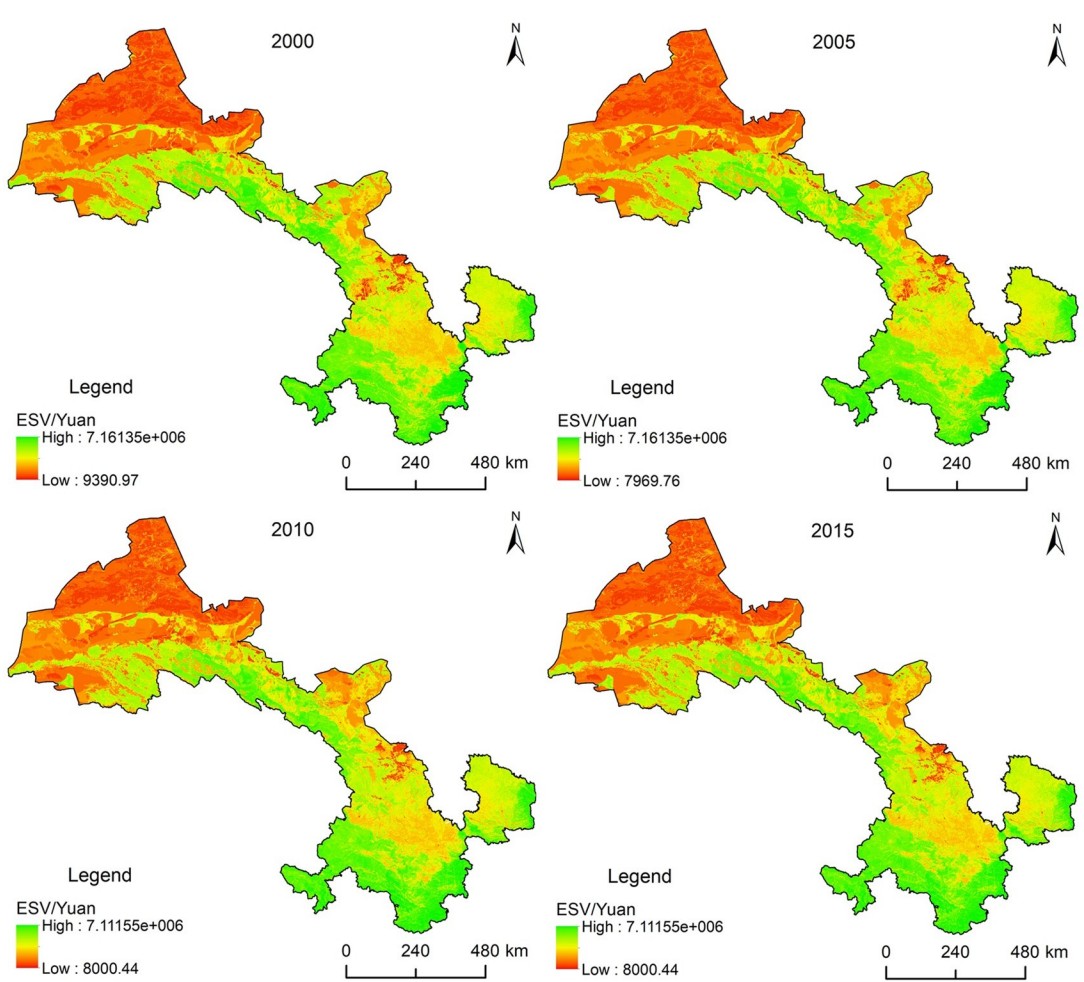

**Fig 4. Ecosystem Service Value (ESV) from 2000 to 2015 in Gansu Province.**

The value of the grassland and forest ecosystems is the highest, accounting for more than 75% of the total value (Table 5), whereas the value of urban and glacial ecosystems is the lowest. In terms of change in each ecosystem type value, the value of forest and urban ecosystems

**Table 4. ESV from 2000 to 2015 in Gansu Province ($10^8$ yuan).**

| ES | 2000 | 2005 | 2010 | 2015 |
|---|---|---|---|---|
| Crops supply | 965.67 | 958.54 | 886.63 | 879.29 |
| Livestock supply | 92.81 | 93.78 | 93.03 | 93.09 |
| Fresh water supply | 199.34 | 201.60 | 197.55 | 198.90 |
| Local climate regulation | 8216.79 | 8,301.09 | 8,164.14 | 8,171.22 |
| Air quality regulation | 1,961.98 | 1,978.94 | 1,925.38 | 1,928.47 |
| Groundwater recharge | 70.64 | 76.43 | 57.67 | 63.62 |
| Soil conservation | 1,551.19 | 1,559.10 | 1,605.89 | 1,606.29 |
| Windbreak and sand fixation | 728.68 | 734.78 | 772.64 | 774.98 |
| Water purification | 2,932.81 | 2,958.28 | 2,819.21 | 2,824.91 |
| Aesthetic value of entertainment | 425.42 | 427.25 | 418.56 | 418.76 |
| Biodiversity maintenance value | 5,215.96 | 5,278.48 | 5,428.37 | 5,436.01 |
| Total value | 2,2361.29 | 22,568.27 | 22,369.07 | 22,395.55 |

**Table 5. Value of each ecosystem type from 2000 to 2015 in Gansu Province ($10^8$ yuan), and proportion (%).**

| Ecosystem | 2000 | | 2005 | | 2010 | | 2015 | |
|---|---|---|---|---|---|---|---|---|
| | Value | Proportion | Value | Proportion | Value | Proportion | Value | Proportion |
| **Forest** | 7,193.20 | 32.17 | 7,312.00 | 32.40 | 7477.01 | 33.43 | 7,489.26 | 33.44 |
| **Grassland** | 10,222.26 | 45.71 | 10,352.27 | 45.87 | 10,181.71 | 45.52 | 10,188.37 | 45.49 |
| **Wetland** | 371.43 | 1.66 | 383.54 | 1.70 | 348.94 | 1.56 | 359.66 | 1.61 |
| **Cultivated land** | 2,969.03 | 13.28 | 2,918.64 | 12.93 | 2,671.66 | 11.94 | 2,659.78 | 11.88 |
| **Cities** | 109.32 | 0.49 | 118.56 | 0.53 | 131.50 | 0.59 | 140.64 | 0.63 |
| **Desert** | 1,484.10 | 6.64 | 1,471.39 | 6.52 | 1,547.69 | 6.92 | 1,547.25 | 6.91 |
| **Glacial** | 11.95 | 0.05 | 11.86 | 0.05 | 10.55 | 0.05 | 10.59 | 0.05 |
| **Total** | 22,361.29 | 100.00 | 22,568.27 | 100.00 | 22,369.07 | 100.00 | 22,395.55 | 100.00 |

is increasing, whereas the value of cultivated land ecosystem is decreasing. The value of grassland, wetland, and glacier ecosystems fluctuates, and the overall trend is decreasing. In contrast, the value of the desert ecosystem fluctuates, and the overall trend is increasing. Over the past 15 y, the total value of the various ecosystem services has increased by 3.43 billion yuan, and the increase in forest ecosystem value has been the highest, whereas the decrease in grassland ecosystem value has been the highest.

## Analysis of the spatiotemporal change in ESV in Gansu Province

Over the past 15 y, the townships with increased ESV are mainly distributed in the east and south of Gansu Province, and west of the Hexi Corridor. The townships which had a decrease in ESV are located in Qilian Mountain and Gannan Plateau (Fig 5).

Over the past 15 y, the number of townships with increased ESV has decreased. There were 959 townships with an increased ESV in 2000–2005, 758 townships in 2005–2010, and only 391 townships increased in value in 2010–2015. From 2000 to 2015, there were 818 townships with increased ESV.

From 2000 to 2005, townships with increased ESV were mainly distributed in the Qilian Mountain, and the eastern and southern parts of Gansu (Tianshui, Pingliang, Qingyang, and Longnan). From 2005 to 2010, the ESV of most townships decreased in the Gannan Plateau and Qilian Mountain, whereas the increased townships were mainly located in Tianshui, Longnan, and the western section of the Hexi Corridor. From 2010 to 2015, the ESV of most townships declined in Lanzhou, Baiyin, Dingxi, Gannan Plateau, and Hexi Corridor, and the townships where the value increased were concentrated in the north and south.

## Discussion

### Advanced value evaluation model

In this study, ESV was evaluated by improving the value equivalent factor per unit area and constructing a regional differential value assessment model in Gansu Province. Different ecosystem types provide different ES types to humans. In the current research on ESV, scholars mostly refer to the classification of ecosystem types by Costanza et al. [5], which is limited to forest, grassland, farmland, wetland, desert, and river. Xie et al. [3] improved this method by accounting for the value of 14 types of ecosystem services in China, and more precisely reflected the differences between ecosystem types. However, this method was based on national scale, and did not meet the needs of practical research, namely a reduction of the research scale and refinement of the classification of ecosystems. Based on the actual ecological situation in Gansu Province, this study identified 7 types of primary ecosystems and 21 types

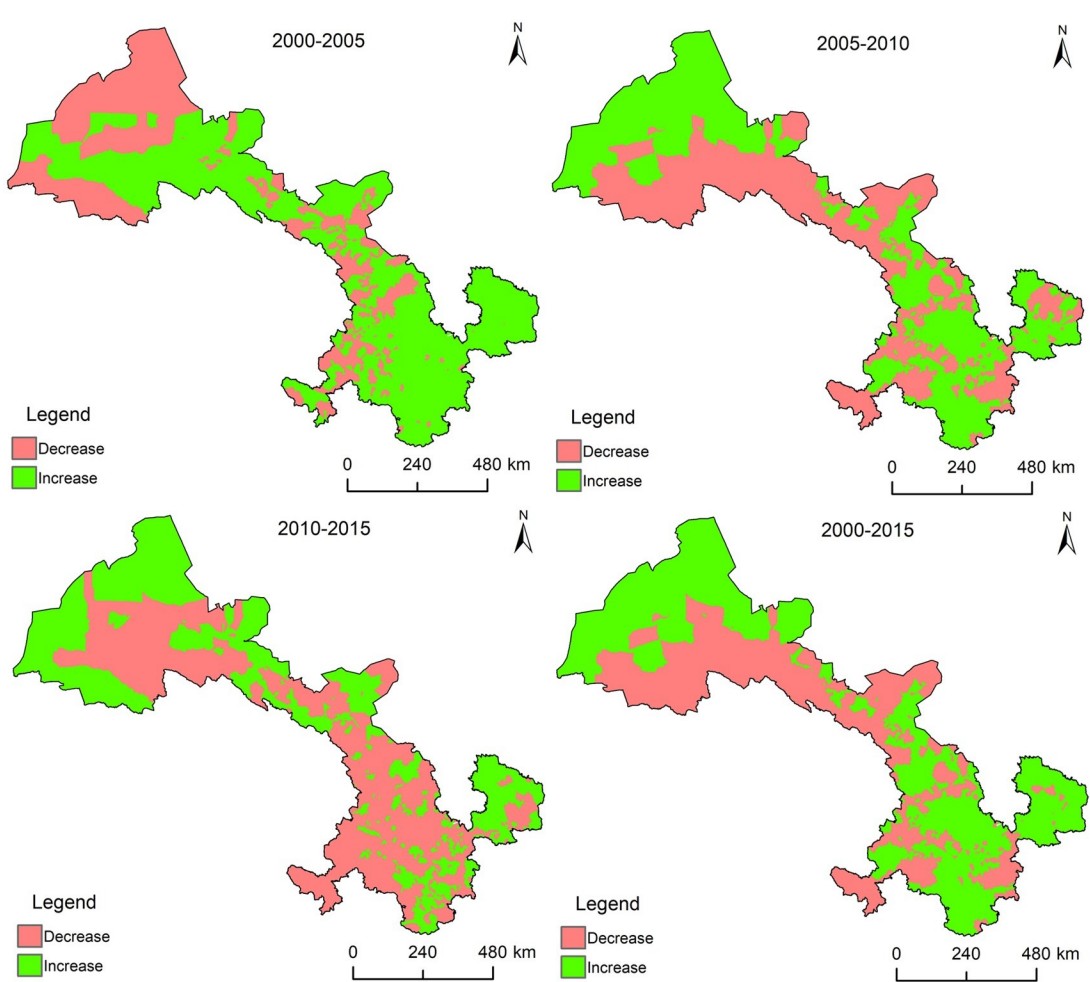

**Fig 5. ESV change distribution map for each town from 2000 to 2015 in Gansu Province.**

of secondary ecosystems to cover the main ecosystem types in Gansu Province more comprehensively. This reflected the differences between the types of ecosystems, and highlighted the importance of ESV.

In different regions, the same ecosystem type provides different ES and their value to humans are quite different. Therefore, the value equivalent factor of Costanza et al. [5] and Xie et al. [3] has been improved by scholars by introducing factors that reflect regional differences, such as NPP, biomass, vegetation coverage, and soil erosion. However, these factors can only reflect regional differences in the main types of ES functions, which are mainly driven by natural factors. It is even more difficult to truly reflect differences in ecological service functions in regions with complex ecological environment characteristics and socioeconomic conditions, such as Gansu Province. In this study, first, the correction index of the value equivalent factor per unit area was constructed. We converted the equivalent factors on the national average level provided by Xie et al. [3] to an average level in Gansu Province, calculated the average level of some of the equivalent factors in Gansu Province, and then converted the equivalent factors in the average level in Gansu Province to a level with significant regional differences by constructing the regional difference adjustment index of each ES function type. In the process of regional difference conversion, the differences in ecological environment quality, resource

endowment, and economic and social development of the different regions were fully considered, thereby more accurately reflecting the ecological well-being of residents in different regions, and the degree of damage to the local ecological environment and resources.

## Errors in the value evaluation model

The accurate construction of the equivalent factor table is the core of the equivalent factor method. In the process of improving the equivalent factor table of Xie et al. [3], this study integrated evaluation results from the literature based on the physical quantity method in Gansu Province or other regions in China. This can avoid or reduce subjective conjecture, which is easily caused by empiricism from the past. The accuracy of the research results in the literature on different ES types affects the size of the equivalent factor in this study. Considering the lack of complete research results on the different ecosystems or ecological service functions in Gansu Province, the ecological service functions of some ecosystem types are investigated based on available research results from other regions in China. This has led to a certain amount of uncertainty in this study. Future research needs to quantitatively calculate the physical quantity of ecosystem types or ecological service functions that are not currently available in the literature, and then determine the equivalent factor to improve the results of this study.

## Reliability of value evaluation models

ESV assessment has been studied extensively in other regions of China, whereas there are few related studies on ESV in Gansu Province. The existing research focuses on the value of a single ecosystem service on a provincial scale [29,34,81], and the calculation method of unit area value is the main method. Research on provincial integrated ESV is almost non-existent at present [83]. The value of forest ecosystem services was 747.70 billion yuan in this study. However, this study does not consider the difference in people's willingness to pay, and their ability to pay, for ESV caused by social development. By including the willingness and ability to pay, the value of forest ecosystem services was 1,971.23 billion yuan in 2010, which is closer to the official release of service value of the forest ecosystem (2,007.97 billion yuan in 2011), and the service value of the forest ecosystem assessed by Wang et al. [27] (2,163.86 billion yuan in 2009) and Wang et al. [82] (1,802.37 billion yuan in 2008). There was, however, a big difference between the ESV evaluated in Gansu Province by Qi [83] and the ESV in this study. The value per unit area was only 98.57 billion yuan in 2010 [83] by using the value calculation method per unit area, resulting in a significantly smaller value.

## Conclusions

Based on the characteristics of ecosystems in Gansu Province, this study developed a revised index based on an increase in some ecosystem equivalent factors, and revised the equivalent factors studied by Xie et al. [3] to form an equivalent factor table in line with ecosystem service valuation in Gansu Province specifically. Eleven regional difference adjustment indices to readjust the value of different service functions were then constructed. The regional difference assessment model constructed in this study distinguished the regional differences in similar ecosystem services to evaluate the ESV in Gansu Province more objectively. The main conclusions are as follows:

1. The desert ecosystem type covers the largest area in Gansu Province, followed by grassland, arable land, and forest ecosystems, and the remaining ecosystems account for only a small part. Desert is mostly located in northern Gansu. Grasslands are mainly located in the Gannan Plateau, Longzhong, and Longdong. Cultivated land is mainly located in Longzhong

and Hexi Corridor, and forests are mainly located in Longnan, Ziwuling, and the Qilian Mountains.

2. From 2000 to 2015, the grassland ecosystem area increased the most, whereas the cultivated land ecosystem area decreased the most. Forest, grassland, and urban ecosystems in Gansu Province continue to increase. Cultivated land ecosystems continue to decrease. Although wetlands and glacial ecosystems have increased over the past 15 y, they have decreased over the long term. The number of desert ecosystems has increased.

3. In 2015, the total ESV for Gansu Province reached 2,239.56 billion yuan, to which the forest and grassland ecosystems contributed the most. In terms of the value of each service function of the ecosystem, the local climate regulation and biodiversity maintenance functions are the main service functions in the province. Regarding the spatial distribution of service values, ESV gradually increases from northeast to southwest, and the areas with high ESV concentrations are Qilian and Longnan Mountains.

4. From 2000 to 2015, ESV increased by 3.43 billion yuan in Gansu Province. The value of forest ecosystems increased the most, whereas the value of grassland ecosystems decreased the most, showing a trend of increasing first, then decreasing, and then slowly increasing again. The value of forest and urban ecosystems continues to increase, and the value of cultivated land ecosystems continues to decrease. From the spatial characteristics of ESV changes, areas with reduced values gradually move from central Gansu to the surrounding areas.

## Supporting information

**S1 Data.**
(ZIP)

## Acknowledgments

We are also grateful for the comments and criticisms of an early version of this manuscript by our colleagues and the journal's reviewers. Acknowledgement for the data support from "National Earth System Science Data Center, National Science & Technology Infrastructure of China. (http://www.geodata.cn)".We would like to thank Editage (www.editage.cn) for English language editing.

## Author Contributions

**Funding acquisition:** Jian Wang.

**Methodology:** Xiaojiong Zhao.

**Project administration:** Jian Wang.

**Validation:** Junde Su.

**Writing – original draft:** Xiaojiong Zhao, Junde Su.

**Writing – review & editing:** Wei Sun.

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
