## [Decision Letter · Decision Letter 0]

27 Oct 2020

PONE-D-20-28811

Evaluation method of ecosystem service value under complex ecological environment: A case study of Gansu Province, China

PLOS ONE

Dear Dr. zhao,

Thank you for submitting your manuscript to PLOS ONE. After careful consideration, we feel that it has merit but does not fully meet PLOS ONE’s publication criteria as it currently stands. Therefore, we invite you to submit a revised version of the manuscript that addresses the points raised during the review process.

We look forward to receiving your revised manuscript.

Kind regards,

Bing Xue, Ph.D.

Academic Editor

PLOS ONE

Journal Requirements:

"The full name of each funder"

4. We note that Figures 1, 3 and 4 in your submission contain map images which may be copyrighted. All PLOS content is published under the Creative Commons Attribution License (CC BY 4.0), which means that the manuscript, images, and Supporting Information files will be freely available online, and any third party is permitted to access, download, copy, distribute, and use these materials in any way, even commercially, with proper attribution. For these reasons, we cannot publish previously copyrighted maps or satellite images created using proprietary data, such as Google software (Google Maps, Street View, and Earth). For more information, see our copyright guidelines: http://journals.plos.org/plosone/s/licenses-and-copyright.

4.1.    You may seek permission from the original copyright holder of Figures 1, 3 and 4 to publish the content specifically under the CC BY 4.0 license. 

4.2.    If you are unable to obtain permission from the original copyright holder to publish these figures under the CC BY 4.0 license or if the copyright holder’s requirements are incompatible with the CC BY 4.0 license, please either i) remove the figure or ii) supply a replacement figure that complies with the CC BY 4.0 license. Please check copyright information on all replacement figures and update the figure caption with source information. If applicable, please specify in the figure caption text when a figure is similar but not identical to the original image and is therefore for illustrative purposes only.

Reviewers' comments:

Reviewer's Responses to Questions

**Comments to the Author**

1. Is the manuscript technically sound, and do the data support the conclusions?

Reviewer #1: Yes

Reviewer #2: Yes

Reviewer #3: Yes

2. Has the statistical analysis been performed appropriately and rigorously? 

Reviewer #1: Yes

Reviewer #2: Yes

Reviewer #3: N/A

3. Have the authors made all data underlying the findings in their manuscript fully available?

Reviewer #1: Yes

Reviewer #2: Yes

Reviewer #3: Yes

4. Is the manuscript presented in an intelligible fashion and written in standard English?

Reviewer #1: Yes

Reviewer #2: Yes

Reviewer #3: No

5. Review Comments to the Author

Reviewer #1: In this paper, xiaojiong Zhao et al. has taken Gansu Province as an example, the regional division method of ecosystem service function is proposed based on the characteristics of differences among different regions, and the variation trend of ESV in Gansu province from 2000 to 2015 is analyzed, which expands the thinking of regional ecological evaluation. The subject matter of this paper is novel, the Angle is clear, the data analysis process is accurate and careful. However, the manuscript needs revision before it is acceptable for publication. The writing of the paper should be improved, I found some of the text is repetition and some parts are not very fluid. The specific sections are questions that should be answered before accepting and proceeding to publications.

1. In introduction, It is recommended that authors add comparisons with other ecosystem service value studies and introduce the advantages of the approach used in this article;

2. In the discussion section, I suggest comparing with other ecological service value assessment methods in Gansu Province;

3.There are still some problems in English grammar and sentences. Please check and correct them carefully.

Reviewer #2: comments to the Author:

The first time I looked at the title " Evaluation method of ecosystem service value under complex ecological environment: A case study of Gansu Province, China ", I was just wondering to know how would the authors deal with this topic. I was a little impressed and felt that this might be a very interesting manuscript. When I reviewed the manuscript, I didn't feel disappointed. In the manuscript, On the one hand, the average value equivalent factor per unit area of different ecosystems is determined in Gansu Province, with reference to the six calculation process. This method of meta-analysis will avoid or reduce the subjective conjecture which is easy to be caused by relying on experts' experience; On the other, through abundant eco-environmental data, this study established a value evaluation model and completed the evaluation of ecosystem service value under complex ecological environment. In particular, one thing is affirmed that the human activity factors that affect ecosystem service value are fully considered by Author to the evaluation of ecosystem service value, such as carrying capacity, air pollution, over exploitation of groundwater and water pollution etc. The methods and data used in the study are new. The manuscript meets the requirement for acceptation for publication.

My main concerns are as follows:

Introduction

1. The description of Gansu Province may be suitable for the part of study area.

2. There lacks enough literature review on the topic in the study area. The authors may check some references on this topic in last decades.

Materials and Methods

1. What is the basis of the ecosystem type classification?

2. Please supplement the distribution map of ecosystem types

3. from 2000-2015, what does this mean? The authors have to state clearly which year was used.

4. Classification of ecosystem service functions. Please explain the specific problem of repeatability. The description is not fully understood due to the ecological integrity has been evaluated in previous research.

Result

1. In the Table3, the unit of area is km2, while in formula 19, the unit of area is ha.

2. Biodiversity maintenance value regulation index (a11), How the habitat quality index is measured?

3. "invest" should be "InVEST"

Other detail question

English needs proofreading and editing, especially those sentences related to comparison. In addition, there exists many formatting problems in this manuscript, please verify the full text.

1.Please unify the number of decimal points in the whole text.

2. Table1. Please note that the black space.

Reviewer #3: (1)The paper takes Gansu Province as an example, on the basis of fully considering the 6 regional differences of ecosystem service function. But it don’t explain why Gansu was taken as an example? Does Gansu have a typical ecological service system?

(2)making dialogue with the ESV evaluation and calculation literatures by taking the role of your specific context.

(3)This paper did a poor job in readability, with many typo errors and unstructured sentences. The authors need a copy editor before next submission.

6. PLOS authors have the option to publish the peer review history of their article (what does this mean?). If published, this will include your full peer review and any attached files.

Reviewer #1: No

Reviewer #2: No

Reviewer #3: No

---

## [Author Response · Author response to Decision Letter 0]

12 Dec 2020

Reply to Reviewer #1

Comment1:

“In introduction, It is recommended that authors add comparisons with other ecosystem service value studies and introduce the advantages of the approach used in this article.”

Response1: 

We added comparisons with other ecosystem service value studies and introduce the advantages of the approach used in this article, line108-118 in marked-up copy of my manuscript. The modified part is highlighted in yellow.

Comment2:

“In the discussion section, I suggest comparing with other ecological service value assessment methods in Gansu Province.”

Response2: 

There are few methods to evaluate the value of ecosystem services in Gansu Province, through literature retrieval, we compared with the study by Wang et al; line735-736 in marked-up copy of my manuscript. The modified part is highlighted in yellow.

Comment3:

“There are still some problems in English grammar and sentences. Please check and correct them carefully.”

Response3: 

We used Editage for English language editing that edited my manuscript, and we have carefully and thoroughly proofread the manuscript to correct all the grammar and typos. For details of manuscript editing, please refer to marked-up copy of my manuscript.

Reply to Reviewer #2

Comment1: 

“In introduction, the description of Gansu Province may be suitable for the part of study area.”

Response1:

In introduction,“Gansu Province is located in the northwestern inland in China. …..Its special geographical location and natural conditions form a more distinct ecological structure.”, this part is repeated with the content in the study area, so this description of Gansu Province has been deleted in introduction. line 119 in marked-up copy of my manuscript. The modified part is highlighted in yellow.

Comment2: 

“There lacks enough literature review on the topic in the study area. The authors may check some references on this topic in last decades. ”

Response2: 

We checked some references about method of ecosystem services value, and added other ecosystem service value studies. line 108-118 in marked-up copy of my manuscript. The modified part is highlighted in yellow.

Comment3: 

“What is the basis of the ecosystem type classification? ”

Response3: 

The basis of the ecosystem type classification was added in Ecosystem type data, line 160-184 in marked-up copy of my manuscript. The modified part is highlighted in yellow.

Comment4:

 “Please supplement the distribution map of ecosystem types.”

Response4:

The distribution map of ecosystem types was added in line 185(Figure 2) in marked-up copy of my manuscript. The modified part is highlighted in yellow.

Comment5: 

“from 2000-2015, what does this mean? The authors have to state clearly which year was used. ”

Response5: 

The mean is “from 2000, 2005, 2010, and 2015”. line 194-195 in marked-up copy of my manuscript. The modified part is highlighted in yellow.

Comment6: 

“Classification of ecosystem service functions. Please explain the specific problem of repeatability. The description is not fully understood due to the ecological integrity has been evaluated in previous research. ”

Response6: 

“The specific problem of repeatability” means is “the double- counting problem between ecological integrity and other ecosystem services”. line 217-219 in marked-up copy of my manuscript. The modified part is highlighted in yellow.

Comment7: 

“In the Table3, the unit of area is km2, while in formula 19, the unit of area is ha. ”

Response7: 

The unit of area is unified as km2. line 593 in marked-up copy of my manuscript. The modified part is highlighted in yellow.

Comment8: 

“Biodiversity maintenance value regulation index (a11), How the habitat quality index is measured? ”

Response8: 

The habitat quality index was measured in detail in another article of the author, so we added the reference. line 562-563 in marked-up copy of my manuscript. The modified part is highlighted in yellow.

Comment9:

“"invest" should be "InVEST"”

Response9: 

We modified the invest to InVEST. line 562 in marked-up copy of my manuscript. The modified part is highlighted in yellow.

Comment10:

“English needs proofreading and editing, especially those sentences related to comparison. In addition, there exists many formatting problems in this manuscript, please verify the full text. ”

Response10: 

We used Editage for English language editing that edited my manuscript, and we have carefully and thoroughly proofread the manuscript to correct all the grammar and typos. For details of manuscript editing, please refer to marked-up copy of my manuscript.

Comment11:

“Please unify the number of decimal points in the whole text. ”

Response11: 

We unified the number of decimal points in the whole text.line 37, 42,567,612,632,646 in marked-up copy of my manuscript. The modified part is highlighted in yellow.

Comment 12:

“Table1. Please note that the black space. ”

Response12: 

We revised the black space. line 314 in marked-up copy of my manuscript. The modified part is highlighted in yellow.

Reply to Reviewer #3

Comment1:

 “The paper takes Gansu Province as an example, on the basis of fully considering the 6 regional differences of ecosystem service function. But it don’t explain why Gansu was taken as an example? Does Gansu have a typical ecological service system? ”

Response1: 

We stated the reasons that the types of ecosystems are complex and diverse in Gansu Province, line 119-127 in marked-up copy of my manuscript. The modified part is highlighted in yellow.

Comment2:

 “making dialogue with the ESV evaluation and calculation literatures by taking the role of your specific context. ”

Response2: 

We have supplemented the relevant references on the other ecosystem service value studies, line108-118, 858-863 in marked-up copy of my manuscript. The modified part is highlighted in yellow.

Comment3:

 “This paper did a poor job in readability, with many typo errors and unstructured sentences. The authors need a copy editor before next submission. ”

Response3: 

We used Editage for English language editing that edited my manuscript, and we have carefully and thoroughly proofread the manuscript to correct all the grammar and typos. For details of manuscript editing, please refer to marked-up copy of my manuscript.

---

## [Decision Letter · Decision Letter 1]

18 Jan 2021

Ecosystem service value evaluation method in a complex ecological environment: A case study of Gansu Province, China

PONE-D-20-28811R1

Dear Dr. zhao,

We’re pleased to inform you that your manuscript has been judged scientifically suitable for publication and will be formally accepted for publication once it meets all outstanding technical requirements.

Kind regards,

Bing Xue, Ph.D.

Academic Editor

PLOS ONE

Additional Editor Comments (optional):

Reviewers' comments:

Reviewer's Responses to Questions

**Comments to the Author**

1. If the authors have adequately addressed your comments raised in a previous round of review and you feel that this manuscript is now acceptable for publication, you may indicate that here to bypass the “Comments to the Author” section, enter your conflict of interest statement in the “Confidential to Editor” section, and submit your "Accept" recommendation.

Reviewer #1: All comments have been addressed

Reviewer #3: All comments have been addressed

2. Is the manuscript technically sound, and do the data support the conclusions?

Reviewer #1: Yes

Reviewer #3: Yes

3. Has the statistical analysis been performed appropriately and rigorously? 

Reviewer #1: Yes

Reviewer #3: Yes

4. Have the authors made all data underlying the findings in their manuscript fully available?

Reviewer #1: Yes

Reviewer #3: Yes

5. Is the manuscript presented in an intelligible fashion and written in standard English?

Reviewer #1: Yes

Reviewer #3: Yes

6. Review Comments to the Author

Reviewer #1: The author has added comparisons with other ecosystem service value studies and introduce the advantages of the approach used in the introduction and grammar problems also have been solved.The overall structure of the article is clear and has strong scientific and practical significance.

Reviewer #3: I really appreciate the revision made by the author, which can be seen to improve the readability of the paper.

7. PLOS authors have the option to publish the peer review history of their article (what does this mean?). If published, this will include your full peer review and any attached files.

Reviewer #1: No

Reviewer #3: No

---

## [Editor Report · Acceptance letter]

21 Jan 2021

PONE-D-20-28811R1 

Ecosystem service value evaluation method in a complex ecological environment: A case study of Gansu Province, China 

Dear Dr. Zhao:

I'm pleased to inform you that your manuscript has been deemed suitable for publication in PLOS ONE. Congratulations! Your manuscript is now with our production department. 

Kind regards, 

on behalf of

Professor Bing Xue 

Academic Editor

PLOS ONE